# Prediction of Consolidation Tumor Ratio on Planning CT Images of Lung Cancer Patients Treated with Radiotherapy Based on Deep Learning

Yizhi Tong [1], Hidetaka Arimura [2,*], Tadamasa Yoshitake [3,*], Yunhao Cui [1], Takumi Kodama [1], Yoshiyuki Shioyama [4], Ronnie Wirestam [5] and Hidetake Yabuuchi [2]

[1] Department of Health Sciences, Graduate School of Medical Sciences, Kyushu University, 3-1-1, Maidashi, Higashi-ku, Fukuoka 812-8582, Japan

[2] Department of Health Sciences, Faculty of Medical Sciences, Kyushu University, 3-1-1, Maidashi, Higashi-ku, Fukuoka 812-8582, Japan; yabuuchi.hidetake.237@m.kyushu-u.ac.jp

[3] Department of Clinical Radiology, Graduate School of Medical Sciences, Kyushu University, 3-1-1, Maidashi, Higashi-ku, Fukuoka 812-8582, Japan

[4] Ion Beam Therapy Center, SAGA HIMAT Foundation, 3049 Harakogamachi, Tosu-shi 841-0071, Japan; shioyama-yoshiyuki@saga-himat.jp

[5] Department of Medical Radiation Physics, Lund University, SE-221 85 Lund, Sweden; ronnie.wirestam@med.lu.se

[*] Correspondence: arimura.hidetaka.616@m.kyushu-u.ac.jp (H.A.); yoshitake.tadamasa.386@m.kyushu-u.ac.jp (T.Y.)

**Abstract:** This study aimed to propose an automated prediction approach of the consolidation tumor ratios (CTRs) of part-solid tumors of patients treated with radiotherapy on treatment planning computed tomography images using deep learning segmentation (DLS) models. For training the DLS model for cancer regions, a total of 115 patients with non-small cell lung cancer (NSCLC) who underwent stereotactic body radiation therapy were selected as the training dataset, including solid, part-solid, and ground-glass opacity tumors. For testing the automated prediction approach of CTRs based on segmented tumor regions, 38 patients with part-solid tumors were selected as an internal test dataset A (IN) from a same institute as the training dataset, and 49 patients as an external test dataset (EX) from a public database. The CTRs for part-solid tumors were predicted as ratios of the maximum diameters of solid components to those of whole tumors. Pearson correlations between reference and predicted CTRs for the two test datasets were 0.953 (IN) and 0.926 (EX) for one of the DLS models ($p < 0.01$). Intraclass correlation coefficients between reference and predicted CTRs for the two test datasets were 0.943 (IN) and 0.904 (EX) for the same DLS models. The findings suggest that the automated prediction approach could be robust in calculating the CTRs of part-solid tumors.

**Keywords:** consolidation tumor ratio; deep learning; part-solid tumors; independent test; non-small cell lung cancer (NSCLC)

## 1. Introduction

Lung cancer is the leading cause of cancer-associated deaths in the United States [1] and Japan [2]. Non-small cell lung cancer (NSCLC), including squamous cell carcinoma (SCC) and adenocarcinoma (ADC), accounts for about 80% of all lung cancer cases [3]. Some factors, e.g., tumor size [4], operability [5], and consolidation tumor ratio (CTR) [6], for NSCLC have been proven to have relationships with treatment outcomes. The CTR is defined as the ratio of the maximum diameter of consolidation (C) to that of a whole tumor (T) on computed tomography (CT). The consolidation means solid components of part-solid tumors. The CTR has been reported to be associated with the prognoses of lung cancer patients who underwent surgical treatment [6–10]. In addition, some other studies

suggested the feasibility of the CTR to predict patients' prognoses after stereotactic body radiation therapy (SBRT) [11,12].

To estimate the CTR in general, clinical physicians need to manually delineate both whole tumor regions and solid components in part-solid tumors [13]. Due to the different levels of experience and skill among physicians, manual contouring may lead to inter-observer and intra-observer variability in contours [14], resulting in uncertainties with random and/or systematic errors in estimating CTRs. To reduce these types of uncertainties, automated prediction approaches for CTRs for lung cancer patients treated with surgical resection have been developed based on deep learning (DL) [15–17]. Sun et al. [15] introduced a DL-based CTR measuring approach for the prediction of tumor invasiveness for lung adenocarcinoma. Wang et al. [16] assessed the invasiveness of lung cancer using a DL-based method for the prediction of 2D and 3D CTRs. Zhu et al. [17] evaluated the performance of a DL algorithm for CTR measurements in predicting the prognosis of lung adenocarcinoma. However, there have been no studies on automated prediction approaches of CTRs on treatment planning CT images for patients who underwent radiotherapy. In addition, the three studies did not assess the segmentation accuracy of their DL models with evaluation indices [e.g., Dice's similarity coefficient (DSC)].

For the prediction of accurate CTRs, automated segmentation techniques for part-solid tumors of lung cancer are necessary. Cui et al. [18,19] investigated a deep learning segmentation (DLS) approach using a dense V-Net for SBRT cases with the three types of tumors [solid, part-solid, and ground-glass opacity tumors (GGO)] on treatment planning CT images. They reported a mean DSC of 0.822 for part-solid tumors [18]. The dense V-Net has shown the advantage for segmentation of small tumors including part-solid tumors [18,19]. Therefore, the dense V-Net could be feasible for the segmentation of whole tumors and measurement of CTRs.

The authors hypothesized that DLS approaches based on the dense V-Net can provide more accurate contours of whole tumors to measure CTRs on treatment planning CT images. In this study, an automated prediction approach of CTRs was proposed for part-solid tumors of patients treated with radiotherapy on treatment planning CT images using a DLS model based on the dense V-Net.

## 2. Materials and Methods

### 2.1. Clinical Cases

This study was approved by the Institutional Review Board of our hospital. For building and testing DLS models, 257 patients were selected with stage I NSCLC in total. The dataset with treatment planning CT images for 208 cases consists of 115 training cases from our hospital and 93 internal test cases (internal test dataset A) from the same hospital. Another dataset included 49 cases selected from NSCLC Radiogenomics [20], the Lung Adenocarcinoma Collection [21] and the Lung Squamous Cell Carcinoma Collection [22] in The Cancer Imaging Archive (TCIA) public dataset, as shown in Table 1. The 208 cases underwent SBRT. The 49 external test cases received radiation therapy or chemo-radiotherapy. The datasets included three types of tumors, i.e., 118 solid, 118 part-solid, and 21 pure GGO tumors, because DLS models have the potential to segment part-solid tumors including GGO and solid components by learning pure GGO and solid tumors as well as part-solid tumors [18]. The internal test dataset B and external test datasets shown in Table 2, which contain only part-solid cases, were employed in the prediction of CTRs. The internal test dataset B includes part-solid tumor cases out of the internal test dataset A. The patients' times to progression were obtained by a radiation oncologist (T.Y.) in a Kaplan–Meier (KM) analysis.

**Table 1.** Patient characteristics with imaging parameters for training, internal test A and external test datasets for building and testing deep learning segmentation models of three lung tumor types.

| Characteristics | Training Dataset (n = 115) | Internal Test Dataset A (n = 93) | External Test Dataset (n = 49) |
|---|---|---|---|
| Institution | Kyushu University Hospital | Kyushu University Hospital | TCIA * |
| Age (years) | 40–92 (median: 76) | 44–89 (median: 78) | 50–85 (median: 72) |
| Sex | | | |
| Male | 69 | 61 | 24 |
| Female | 46 | 32 | 25 |
| Tumor type | | | |
| Solid | 73 | 45 | 0 |
| Part-solid | 31 | 38 | 49 |
| Pure GGO | 11 | 10 | 0 |
| Matrix size | 512 × 512 | 512 × 512 | 512 × 512 |
| Number of slices | 103–235 | 103–225 | 38–515 |
| Pixel size (mm) | 0.78–0.98 | 0.78–0.98 | 0.55–0.98 |
| Slice thickness (mm) | 2.0 | 2.0, 3.2 | 0.63–5.0 |

* TCIA: The Cancer Imaging Archive.

**Table 2.** Patient characteristics with imaging parameters for internal test dataset B and external test dataset for prediction of CTRs for part-solid tumor cases.

| Characteristics | Internal Test Dataset B (n = 38) | External Test Dataset (n = 49) |
|---|---|---|
| Age | 59–89 (median: 79.5) | 50–85 (median: 72) |
| Sex | | |
| Male | 22 | 24 |
| Female | 16 | 25 |
| Tumor type | | |
| Part-solid | 38 | 49 |
| Matrix size | 512 × 512 | 512 × 512 |
| Number of slices | 120–224 | 38–515 |
| Pixel size (mm) | 0.78–0.98 | 0.55–0.98 |
| Slice thickness (mm) | 2.0 | 0.625–5.0 |
| Histology | | |
| Adenocarcinoma | 17 | 39 |
| Squamous cell carcinoma | 1 | 10 |
| Unknown | 20 | 0 |
| Time to progression | | |
| Unknown | 16 | 0 |
| Range (days) | 283–2329 (median: 920) | 12–2793 (median: 1014) |
| Maximum diameters of solid components (mm) * | 1.5–57.5 (median: 20.3) | 1.5–81.7 (median: 32.6) |
| Maximum diameters of whole tumors (mm) * | 10.5–67.3 (median: 32.3) | 21.4–81.7 (median: 42.4) |

* Measurement based on an iso-voxel size of 1.5 mm.

Patients from our hospital were scanned while breathing freely to acquire the planning CT images using 4-slice CT scanners (Mx 8000, Philips, Amsterdam, The Netherlands; Aquilion, Toshiba, Tokyo, Japan; Aquilion PRIME, Toshiba, Tokyo, Japan). Patients from TCIA were scanned with CT scanners from Siemens and GE. A radiologist (H.Y.) determined the three tumor types based on the planning CT images with a lung window level (WL) of −600 and window width (WW) of 1500 Hounsfield units (HU) [23]. All lung cancer regions on CT images were observed by the radiologist with almost the same contrast and brightness, which were manually adjusted according to the WL and WW. Since CT images are obtained from invariant physical values (linear attenuation coefficients) [24], the CT values (HU) including WL and WW could ideally be independent of CT scanners.

### 2.2. Overall Workflow

Figures 1 and 2 show the overall workflows of the developments of the DLS model and automated prediction of CTRs, respectively. The dense V-Net was trained with the training dataset, and the trained model was evaluated with internal test dataset A by using the DSC. Whole tumor regions were predicted for internal test dataset B and the external test dataset. Then, a thresholding technique with a clinical window level and width was applied to the CT images within the predicted whole tumor regions to obtain solid components. Finally, the maximum diameters for both the whole tumor and the part-solid regions were measured by using the Feret method. The CTR for each part-solid tumor was predicted as a ratio of the maximum diameter of the largest solid component to that of the whole tumor. The automated prediction approach of CTRs was assessed with the Pearson correlation coefficient (PCC) [25] and the intraclass correlation coefficient (ICC) [26]. The prognostic power of predicted CTRs was evaluated with *p*-values in a KM analysis.

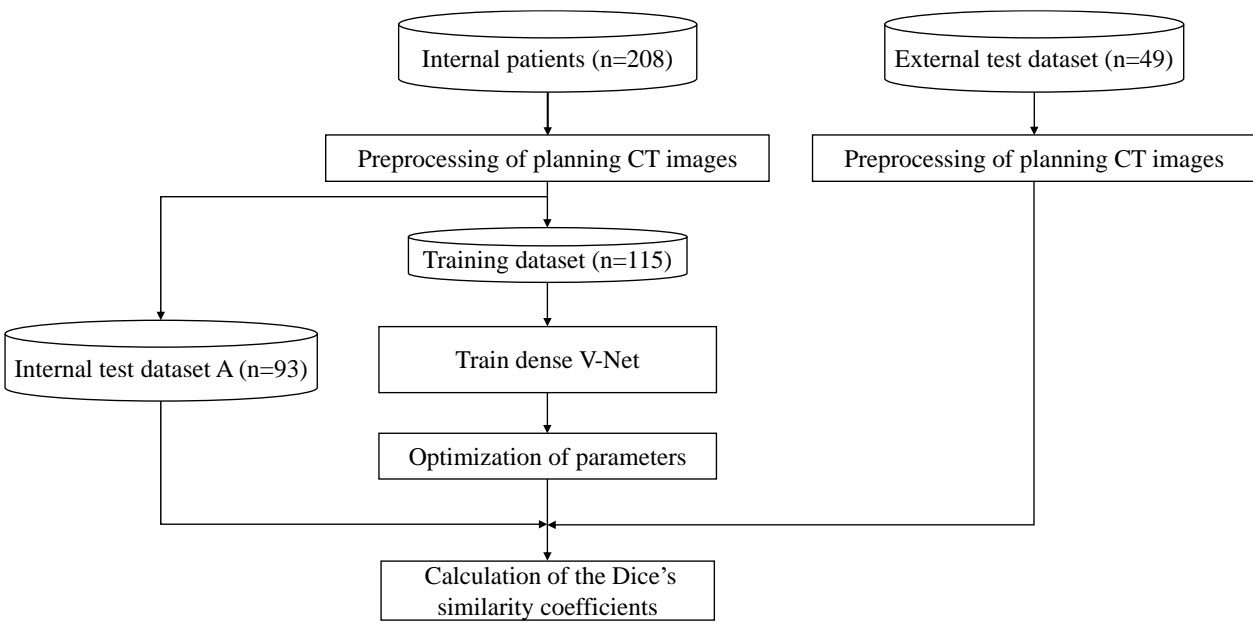

**Figure 1.** The overall workflow for a deep learning (dense V-Net) segmentation (DLS) model of lung cancers.

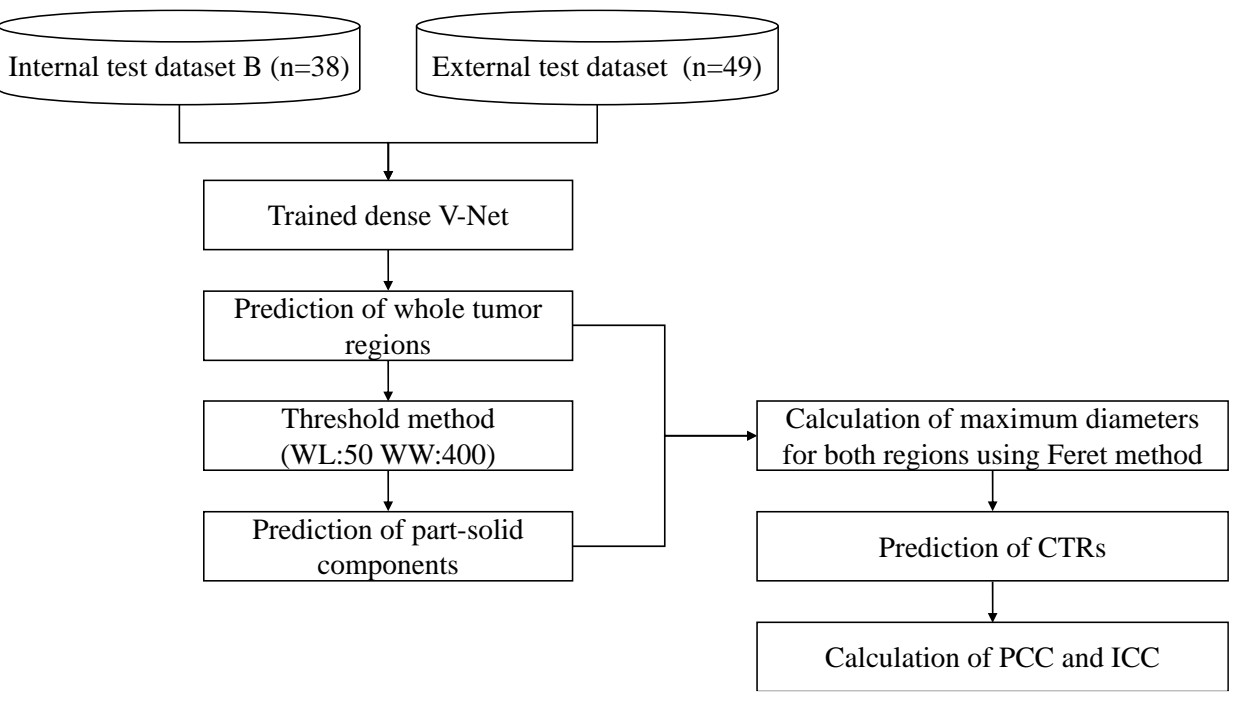

**Figure 2.** The overall workflow for an automated prediction approach of CTRs of part-solid tumor cases. PCC: Pearson correlation coefficient, ICC: intraclass correlation coefficient, WL: window level, WW: window width.

### 2.3. Preprocessing of Dataset

The three-dimensional (3D) treatment planning CT images and gross tumor volume (GTV) regions were converted into iso-voxel images (1.5 mm) using a cubic interpolation method and a shape-based interpolation method [27], respectively. A $40 \times 40$-pixel region centered on the GTV centroid for all axial slices containing each GTV was extracted from the 3D CT image and was inputted into the DLS model.

### 2.4. DLS Model

The dense V-Net [28] was constructed as the DLS model in this research according to the previous study [19]. The architecture of the revised model is illustrated in Figure S1. The hyperparameters of the DLS model are included in Table S2. A Laplacian of Gaussian (LoG) filter was applied to all treatment planning CT images to enhance the edges and reduce image noise. The input CT image size and output tumor region image size for the DLS model was $40 \times 40 \times 40$ pixels. The model outputs were binarized with a threshold of 0.5 to obtain segmented tumor regions. The dilated convolution and spatial prior in the dense V-net were removed since they did not generate significant benefit to the performance [28]. To reduce the calculation time and the number of parameters, a bottleneck structure [29] was used in the densely connected layers. Except for the final layer, the activation function was a Scaled Exponential Linear Unit (SELU) [30], which is the best activation function in [18]. SELU was defined as

$$SELU(x) = s(\max(0, x) + \min(0, \alpha(e^x - 1)))$$ (1)

where $s$ and $\alpha$ were the constant values, which were set as $\alpha = 1.6733$ and $s = 1.0507$, respectively, based on a past study [30].

### 2.5. Prediction of CTR

The CTR for a part-solid tumor was defined as the ratio of the maximum diameter of the largest solid component to the maximum diameter of the whole tumor region. Therefore,

the CTR of a pure GGO is 0, and the CTR of a solid tumor is 1 [9]. An example of the estimation of a solid component (C) and whole tumor region (T) is illustrated in Figure 3.

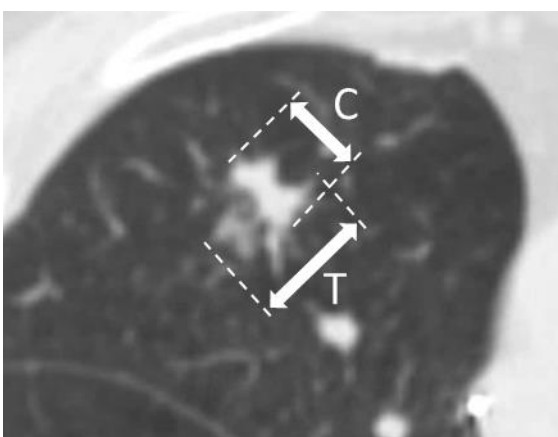

**Figure 3.** A measurement of a part-solid component (C) and a whole tumor region (T) on a planning CT image for measurement of a CTR (C/T).

To extract solid components of a whole tumor, a thresholding technique was applied with a mediastinal window, a WL of 50 HU and WW of 400 HU [23] within a whole tumor region that was obtained from the DLS model. The largest solid component was selected by counting the number of voxels within solid components in a whole tumor region when there are more than two solid components in the tumor region.

*The AJCC Cancer Staging Manual (8th edition)* [31] states that a tumor size should be obtained in its greatest dimension, which may be assessed by the axial, coronal and sagittal directions. If technically possible, all the different projections should be investigated to determine the tumor size. In this study, the 3D Feret method was adopted to calculate the maximum diameters of the whole tumor region and solid component for all projections [32]. The Feret diameter is the distance between two parallel tangents to the projection of the tumor in a certain direction (Figure 4). The thresholding technique and Feret method were applied to both whole tumor regions predicted by DLS models and reference whole tumor regions to determine the predicted and reference CTRs.

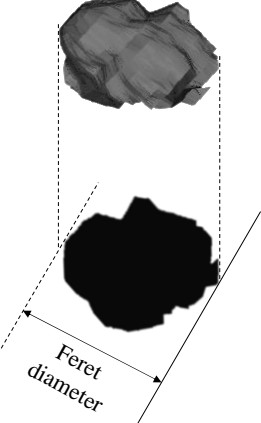

**Figure 4.** An illustration of the Feret diameter, the maximum diameter using the Feret method.

The counting of the voxels depends on one-voxel connectivity, i.e., a 6-connectivity approach [33]. The example for a 3-connectivity approach is shown in Figure S2.

*2.6. Evaluation and Statistical Analysis*

The segmentation accuracies for the DLS model were evaluated using the DSC [34]. All DSCs were calculated from the 3D volumes predicted by the DLS model. The DSC indicates the similarity between predicted contours and reference contours delineated by a radiation oncologist. DSC could be calculated by the following equation:

$$DSC = \frac{2n(T \cap D)}{n(T) + n(D)},$$

where $T$ is the reference contour delineated by a radiation oncologist and $D$ is the contour estimated using the DLS method. $n(T)$ is the number of voxels in contour $T$, $n(D)$ is the number of voxels in contour $D$ and $n(T \cap D)$ is number of overlapping voxels between reference contour $T$ and estimated contour $D$.

The accuracy for the automated prediction approach of CTRs was assessed using the PCC, the ICC and a Bland–Altman analysis [35]. The PCC evaluates the correlation between the reference and predicted CTRs for both the internal (B) and external test datasets. The ICC of ICC (2,1) tests the reliability between the reference and predicted CTRs. ICC (2,1) is the intraclass correlation coefficient where the model is two-way random effects, the type is single rater/measurement, and the definition of relationship considered to be important is absolute agreement [26].

The degree of agreement between reference and predicted CTRs was assessed using the Bland–Altman analysis. The limits of agreement in the analysis were defined as the mean difference $\pm$ 1.96 $\times$ standard deviation of the differences, and systematic biases in the differences between predicted and measured values were evaluated.

Prognostic predictability of predicted CTRs was assessed by using $p$-values in a KM analysis [36], which has been widely used in time survival analyses in the medical field [37]. The $p$-values were evaluated using a log-rank test in KM curves for time to progression between low-risk and high-risk patients, who were stratified with the medians of predicted CTRs for internal test dataset B and the external test dataset. If the CTR of a patient was higher than the median, the patient was considered a high-risk patient. Otherwise, the patient was regarded as a low-risk patient. The $p$-values less than 0.05 were considered statistically significant differences.

*2.7. Implementation of Automated Prediction Approach of CTRs*

The conversion of iso-voxel images, centroid-based cropping and quantization were performed using MATLAB R2018a (MathWorks, Inc., Natick, MA, USA). All other processes were performed in Python 3.8, accelerated by CuPy 11.1, an open-source GPU accelerating library for NumPy and SciPy.

The DLS model was constructed using TensorFlow 2.5.0 and trained with 115 training cases in Table 1. To avoid overfitting when using extremely small training datasets, data augmentation including image flipping, rotation and rescaling was separately applied to the training dataset. The parameters to be optimized were learning rate, batch size, number of iterations, parameters of LoG processing sigma and weight of Log processing beta. The candidate batch sizes were 64, 32, 16 and 8 in this study. To reduce the time cost for optimizing the hyperparameters, the Dice loss was fixed as the loss function [29], which was the best loss function in the previous study [18]. The optimal iteration was defined as the one with the lowest dice loss on the validating data of the training dataset within 9000 iterations. The hyperparameters were optimized separately on a server with two NVIDIA GeForce RTX 3090 devices and a server with two NVIDIA Quadro P6000 devices (NVIDIA Corporation, Los Alamitos, CA, USA). The Feret diameters of whole tumors and solid components were calculated based on a Python package for extracting 2D and 3D shape measurements from images (imea) [38].

## 3. Results

Figure 5 shows DSCs of regions for three types of lung tumors (solid, part-solid, and pure GGO) and all types obtained from the DLS model (dense V-Net) for internal test dataset A. The DSCs for all types of tumors, solid tumors, part-solid tumors and pure GGO tumors are $0.810 \pm 0.08$ (mean $\pm$ SD), $0.828 \pm 0.08$, $0.784 \pm 0.09$ and $0.824 \pm 0.03$. Figure 6 shows DSCs for part-solid tumor regions obtained from the DLS model for internal test dataset B and the external test dataset. The DLS model exhibited DSCs of $0.784 \pm 0.09$ and $0.776 \pm 0.107$ for internal test dataset B and the external test dataset, respectively.

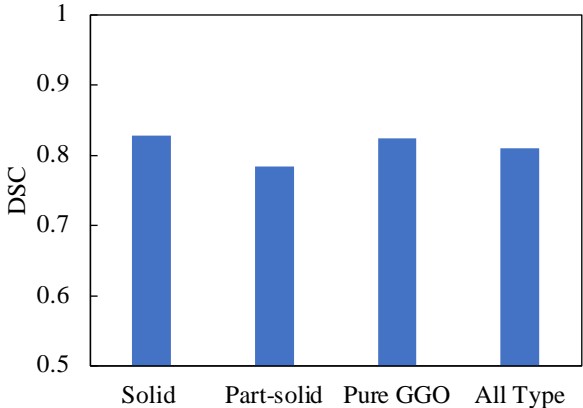

**Figure 5.** DSCs of regions for three types of lung tumors (solid, part solid, and pure GGO) and all types obtained using a DLS model (dense V-Net) for internal test dataset A.

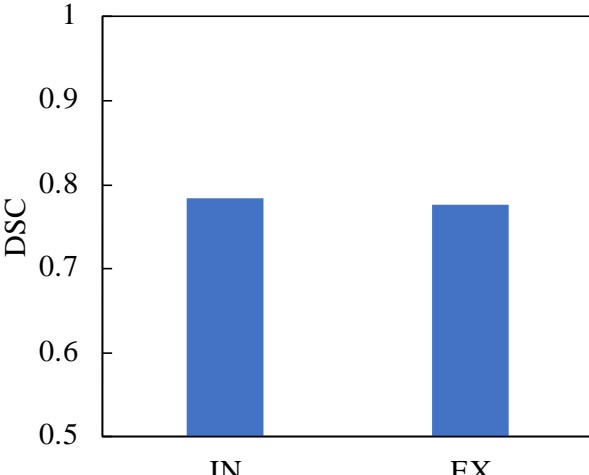

**Figure 6.** DSCs of part-solid tumor regions obtained by a DLS model (dense V-Net) for internal test dataset B (IN) and the external test dataset (EX).

Figure 7 shows the correlation between reference and predicted CTRs with the DLS model for the internal test dataset B. The predicted CTRs showed a PCC of 0.953 and an ICC of 0.943. Figure 8 shows the Bland–Altman plot of agreement between reference and predicted CTRs for the internal test dataset B. The horizontal lines are drawn at the mean difference of 0.03, and 95% of internal test dataset B (36 of 38 cases) is within the limits of agreement. Figure 9 shows four cases that indicated the highest and lowest differences in CTR between reference and predicted CTRs.

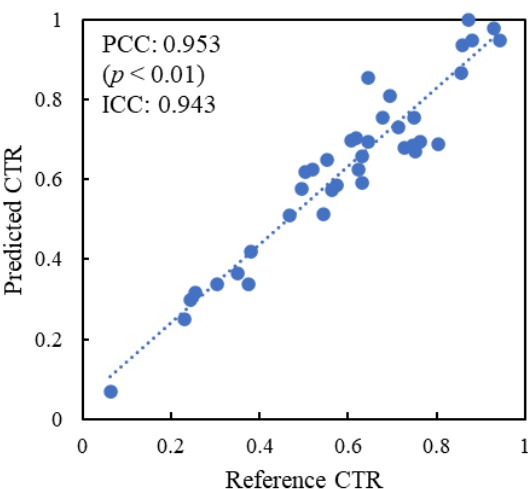

**Figure 7.** Correlation between reference and predicted CTRs with the proposed model for an internal test dataset B. PCC: Pearson correlation coefficient, ICC: intraclass correlation coefficient.

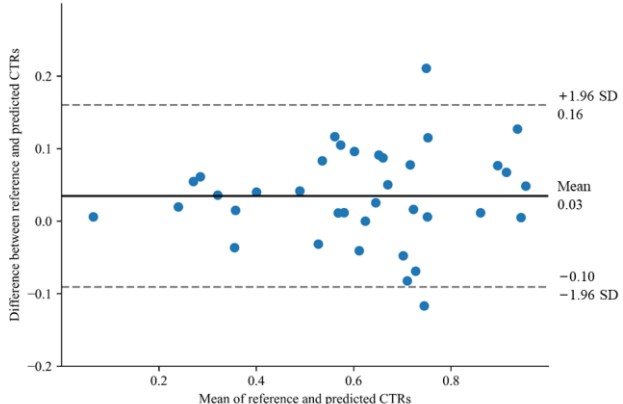

**Figure 8.** Bland–Altman plot of reference and predicted CTRs by the proposed model for an internal test dataset B. SD: standard deviation.

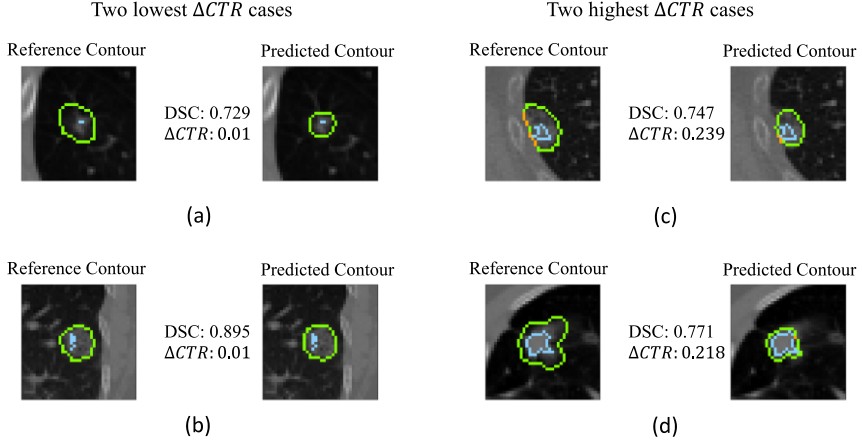

**Figure 9.** Four cases that indicated the lowest (**a,b**) and highest (**c,d**) differences in CTR (ΔCTR) between reference and predicted CTRs by the proposed model for an internal test dataset B. ΔCTR = |Reference CTR–Predicted CTR|. Green line: whole tumor contour, sky blue line: solid component contour, orange line: overlapping contour.

Correlation between reference and predicted maximum diameters of both whole tumor regions and solid components and the Bland–Altman plot of agreement between reference diameters and diameters predicted with the DLS model are shown in Figures S3–S6.

Figure 10 shows the correlation between reference and predicted CTRs with the dense V-Net for the external test dataset. The CTRs predicted by the dense V-Net still show a PCC of 0.926 and an ICC of 0.904. Figure 11 shows the Bland–Altman plot of agreement between reference CTRs and predicted CTRs for the external test dataset. The horizontal lines are drawn at the mean difference of 0.05 (solid line), and 92% of the external test dataset (45 of 49 cases) is within the limits of agreement (dotted lines).

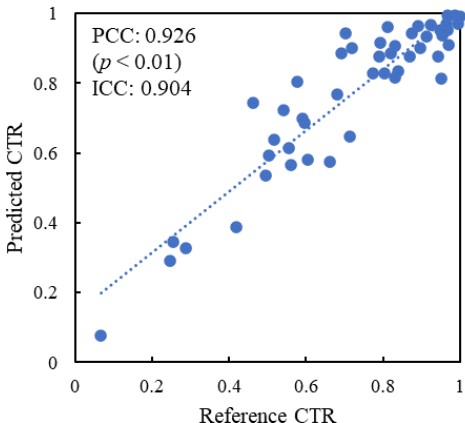

**Figure 10.** Correlation between reference and predicted CTRs with the proposed model for an external test dataset. PCC: Pearson correlation coefficient, ICC: intraclass correlation coefficient.

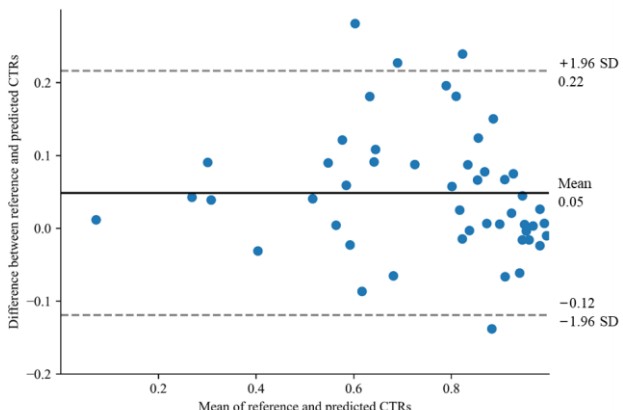

**Figure 11.** Bland–Altman plot of reference and predicted CTRs by the proposed model for an external test dataset. SD: standard deviation.

Correlation between the reference and predicted maximum diameters of both whole tumor regions and solid components and the Bland–Altman plot of agreement between reference diameters and diameters predicted with the proposed model with the dense V-Net are shown in Figures S7–S10.

Figure 12 shows four cases that indicated the highest and lowest differences in CTR between reference and predicted CTRs.

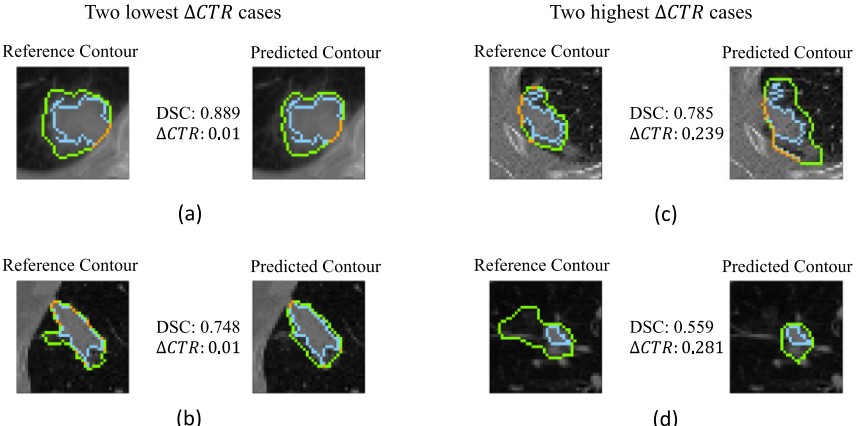

**Figure 12.** Four cases that indicated the lowest (**a,b**) and highest (**c,d**) differences in CTR (ΔCTR) between reference and predicted CTRs by the proposed model for the external test dataset. ΔCTR = |Reference CTR–Predicted CTR|. Green line: whole tumor contour, sky blue line: solid component contour, orange line: overlapping contour.

Figure 13 shows the KM curves for time to progression between low-risk and high-risk patients, who were stratified with the medians of predicted CTRs for internal test dataset B (left) and the external test dataset (right). There were statistically significant differences between the low-risk and high-risk patients.

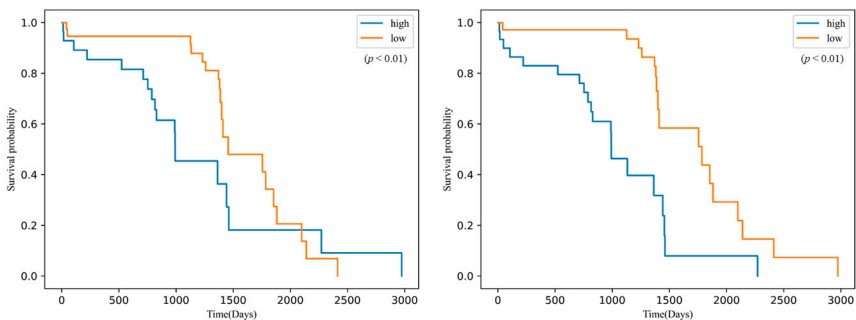

**Figure 13.** Kaplan–Meier survival curves for time to progression between low-risk and high-risk patients who were stratified with the medians of reference CTRs (**left**) and predicted CTRs (**right**) for internal test dataset B and the external test dataset.

## 4. Discussion

The proposed automated prediction approach for CTRs showed high PCCs of 0.953 (IN) and 0.926 (EX) and high ICCs of 0.943 (IN) and 0.904 (EX) for the prediction of CTRs on both internal test dataset B and the external test dataset, respectively. Further to this, the predicted CTRs showed the prognostic power of both datasets in stratifying SBRT patients into low-risk and high-risk patients, as shown in Figure 13.

As shown in Figure 9, the two worst cases (Figure 9c,d) showed lower DSCs of 0.747 and 0.771 than the best case (Figure 9b) (DSC: 0.895). These lower DSCs affected the difference between reference and predicted contours and caused the larger differences in CTRs. The solid component contours are similar to each other between the predicted and reference ones. Since the whole tumor contour varied, this variability of whole tumor contours could cause the larger CTR differences. The other best case (Figure 9a) shows a lower DSC of 0.729 and low ΔCTR because of the small sizes of the solid component and large whole tumor regions. On the other hand, the two best cases (Figure 9a,b) showed higher DSCs of 0.889 and 0.748 than the worst case (Figure 9d) (DSC: 0.559) as shown in Figure 12. The worst case (Figure 12c) shows a DSC of 0.785, however, the tumor edge is

erroneously involved in a tumor region and leads to the lower accuracy. This issue could be solved by combining other DLS models [39] with our approach.

DL-based prediction approaches for CTRs have been investigated for lung cancer patients treated with surgical resection [15–17]. Sun et al. [15] reported an area under receiver operating characteristic curve (AUC) of 0.803 (validation test) for the prediction of tumor invasiveness for lung adenocarcinoma. Wang et al. [16] demonstrated the prediction power of tumor invasiveness with an AUC of 0.826 (validation test). Zhu et al. [17] showed ICCs of 0.838 and 0.832 for DL-based measurements of maximum whole tumor size and maximum solid component size, respectively. In this study, the proposed prediction model exhibited ICCs of 0.980 and 0.801 for both sizes of the external test dataset as shown in Figures S7 and S9.

*The AJCC Cancer Staging Manual (8th edition)* suggested to the radiologists to estimate the tumor sizes in all different projections if it is technically possible [31]. Dinkel et al. [39] proposed a method that can measure the diameters of lung cancer regions on the three orthogonal views (axial, coronal and sagittal views). Haddad et al. [40] measured the diameters of breast cancers by dividing the cancer into serval slices in one dimension, whereas the proposed approach measured the maximum diameters of whole tumor regions and solid components with the Feret method for all projections. Thus, the proposed approach in this study followed the guidelines rather than past studies. Heuvelmans et al. [41] measured the lung tumor diameters by a semi-automatic method based on the contours delineated by two radiologists. On the other hand, the authors developed the DLS model to automatically predict the contours, which could reduce the labors of radiologists, for the prediction of CTRs.

CT image quality (e.g., motion artifacts or image blurring) degraded by respiratory motion could affect the segmentation accuracy of lung tumors [42]. In addition, a review reported that the performance of deep learning-based segmentation approaches has been deteriorated by low image quality such as low contrast objects [43]. The image quality of treatment planning CT images used in this study may be affected by respiratory motion. Even so, in our results, the proposed prediction approach of CTRs reached ICCs of 0.943 (IN) and 0.904 (EX), possibly because the LoG filter for all treatment planning CT images enhanced the edges and reduced image noise. Nevertheless, the image quality of the treatment planning CT images should be improved continuously using other filters, which could reduce motion artifacts [44].

There are several limitations in this study. First, unbalanced training datasets were employed, including 73 solid, 31 part-solid and 11 pure GGO tumors. This may be the reason for the imbalance in performance for the different tumor types and could affect the segmentation accuracy of the pure GGO and part-solid tumors. This could influence the performance of the predicted CTR. Therefore, training the model on a larger number of cases, especially more part-solid tumors, is needed to improve the accuracy for the prediction of CTRs. Although the image augmentation technique with image flipping, rotation and rescaling was applied to increase the training data, more pure GGO and part-solid cases will be collected from Kyushu University Hospital or open-source datasets with prognostic data. In addition, advanced augmentation techniques including generative adversarial networks [45] can be applied to adjust the number of unbalanced data. Second, the DSCs for three types of tumors obtained with our DLS model were still below 0.9, and especially, were lower than 0.8 for part-solid tumors. The performance can be improved by using novel networks, for instance morphological vision transformers [46]. Finally, the 38 cases in the internal test dataset underwent SBRT, but the 49 external test cases received radiation therapy or chemo-radiotherapy. Since the patients in this study received different types of radiotherapy, the times to progression for the KM analysis may have been affected by the treatment methods. Therefore, we should use data from patients who underwent the same type of radiotherapy in future work.

## 5. Conclusions

The authors proposed an automated prediction approach for calculating CTRs of part-solid tumors, which achieved high ICCs of 0.943 and 0.904 between reference and predicted CTRs for the internal and external test datasets, showing its robustness for clinical use. The predicted CTRs showed the prognostic power of both datasets in predicting radiotherapy patients with low-risk and high-risk. Future work will involve investigating the improvement of the DLS model and application in patients treated with the same type of radiotherapy.

**Supplementary Materials:** The following supporting information can be downloaded at: https://www.mdpi.com/article/10.3390/app14083275/s1, Figure S1: Architectures of a deep learning segmentation (DLS) model (dense V-Net); Figure S2: Visualization of 6-connectivity; Figure S3: Correlation between reference and predicted maximum diameters of whole tumor regions with the proposed model for an internal test dataset B. PCC: Pearson correlation coefficient, ICC: Intraclass correlation coefficient; Figure S4: Bland-Altman plot of reference and predicted maximum diameters of whole tumor regions with the proposed model for an internal test dataset B; Figure S5: Correlation between reference and predicted maximum diameters of solid components with the proposed model for an internal test dataset B. PCC: Pearson correlation coefficient, ICC: Intraclass correlation coefficient; Figure S6: Bland-Altman plot of reference and predicted maximum diameters of solid components with the proposed model for an internal test dataset B; Figure S7: Correlation between reference and predicted maximum diameters of whole tumor regions with the proposed model for an external test dataset. PCC: Pearson correlation coefficient, ICC: Intraclass correlation coefficient; Figure S8: Bland-Altman plot of reference and predicted maximum diameters of whole tumor regions with the proposed model for an external test dataset; Figure S9: Correlation between reference and predicted maximum diameters of solid components with the proposed model for an external test dataset. PCC: Pearson correlation coefficient, ICC: Intraclass correlation coefficient; Figure S10: Bland-Altman plot of reference and predicted maximum diameters of solid components with the proposed model for an external test dataset; Table S1: DSCs for three types of lung tumors (solid, part-solid, pure GGO) and all types using dense V-Net for internal test dataset A and external test dataset; Table S2: Hyper-parameters for Dense V-Net.

**Author Contributions:** Conceptualization: H.A. and Y.T.; data curation: Y.S., T.Y., Y.T., Y.C., T.K. and H.A.; formal analysis: Y.T., H.A. and R.W.; funding acquisition: H.A.; investigation: Y.T., Y.C. and H.A.; methodology: Y.T., H.A. and R.W.; project administration: H.A.; resources: Y.T., H.A., T.Y., Y.S. and H.Y.; software: Y.T. and Y.C.; supervision: H.A.; validation: Y.C., H.A., T.Y., Y.S. and H.Y.; visualization: Y.C. and H.A.; writing—original draft: Y.C., H.A. and R.W. All authors have read and agreed to the published version of the manuscript.

**Funding:** This study was partially supported by JST SPRING Grant Number JPMJSP2136 and JSPS KAKENHI Grant (Number: JP24K10840).

**Institutional Review Board Statement:** The study was conducted according to the guidelines of the Declaration of Helsinki and approved by the Institutional Review Board of Kyushu University Hospital (protocol code 2020-472, 2 November 2020).

**Informed Consent Statement:** Patient consent was waived due to the retrospective nature of the study.

**Data Availability Statement:** Restrictions apply to the datasets: the datasets presented in this article are not readily available because they belong to Kyushu University Hospital. Requests to access the datasets should be directed to Kyushu University Hospital.

**Acknowledgments:** The authors are grateful to all the members of the Arimura Laboratory https://web.shs.kyushu-u.ac.jp/~arimura/ (accessed on 9 April 2024), whose comments contributed to this study. Ronnie Wirestam was supported by the Swedish Foundation for International Cooperation in Research and Higher Education (STINT).

**Conflicts of Interest:** The authors declare no conflicts of interest.

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
