# Peer review of "Prediction of Consolidation Tumor Ratio on Planning CT Images of Lung Cancer Patients Treated with Radiotherapy Based on Deep Learning"

_applsci, doi:10.3390/app14083275_

Round 1

Reviewer 1 Report

Comments and Suggestions for Authors

The manuscript proposes an automated prediction method of CTRs of stage I non-small cell lung cancer using deep learning segmentation model with treatment planning CT images.

The design, implementation, and evaluation of the proposed method are well introduced. The readers of the journal can get insight into the research question and the research contribution based on the manuscript.

Some suggestions and comments are listed in the following for the authors to improve the manuscript.

1. In the proposed method, dense V-Net is used. It is highly suggested that the authors can elaborate on the design decision in more detail (e.g., why dense V-Net, any modification to the DL model, and so on).

2. A comprehensive literature review is highly suggested. Thus, readers can get more insights into the status quo and the remaining challenges.

3. Figure S1 and Table S2 mentioned in line 131 and line 132 are unavailable.

4. Based on the manuscript, the dense V-Net is revised in the proposed model. Please elaborate on the design in more detail.

5. Please check the grammar of the sentence in line 152.

6. Figure S2 mentioned in line 158 is unavailable.

7. What is ICC(2,1) in line 177?

8. Figure S3-S6 mentioned in line 212 are unavailable.

9. Figure S7-S10 mentioned in line 240 are unavailable.

10. Please elaborate on the rule of the classification of low-risk and high-risk patients. Please explain how the result of Figure 13 is analyzed and identified.

11. Please revise the conclusion section and give future research directions.

Reviewer 2 Report

Comments and Suggestions for Authors

This paper aims to propose an automatic prediction method for solid tumor ratio (CTR) of stage I non-small cell lung cancer based on deep learning segmentation (DLS) models and computed tomography images of treatment plans. The research results show that this method has shown robust performance in both internal and external test datasets.

The comments of the paper are as follows:

1. Introduction part.

(1) The disadvantages of the existing automated segmentation techniques for lung cancer regions should be introduced in detail. To what extent have the latest research achievements in this field been achieved?

(2) Why deep learning segmentation methods are used to predict CTR and lack of advantages of deep learning segmentation methods. The description of the deep learning segmentation method is insufficient and needs to be extended in more detail.

2. Materials and Methods part.

(1) The description of the revised model in Section 2.4 is not enough, and it is necessary to demonstrate the details of the improved model. Why modify the model? What is the difference between the improved model and other common models?

(2) Lack of description on the training environment and parameters of the DLS model.

(3) The data of cases or sample is insufficient, especially for part-solid tumors. It will affect the accuracy of the prediction of CTRs. What is your solution?

(4) The organizational structure of Section 2 should be rearranged to enhance the readability and clarity of this paper.

3. Results part.

(1) The evaluation of findings is not comprehensive enough.

(2) The research results should be compared with other state-of-art methods, suggest creating a table for explanation. 

Comments on the Quality of English Language

The English expression of this paper is relatively clear. However, minor editing of English language required. 

Reviewer 3 Report

Comments and Suggestions for Authors

The authors of the article “Automated prediction of consolidation tumor ratio for stage I  non-small cell lung cancer from treatment planning CT images based on deep learning segmentation models.” present an interesting contribution to detecting cancer using a diameters ratio.

There are some comments:

1) The acronym NSCLC is not defined on line 23

2) The title of the article is very long. 

3) In Fig 2, it is indicated in the Threshold block, WL:50 WW:400

   What do WL and WW mean? And what do those values wl=50 and ww = 400 mean?

The method you propose is very sensitive to the threshold. Is there a procedure to detect this parameter automatically?

Because they use images from another database with different equipment, how can we ensure the threshold will work for all images?

4) In Fig 3, it is recommended that the region indicated as the tumor and the surrounding region be separated and enlarged to exemplify the better.

5) What software did you use to obtain the Feret diameter?

6) What is the justification for using the Wilcoxon signed-rank test and Kaplan-Meier (KM) analysis?

The graphs and results indicate that the proposal is good.

Reviewer 4 Report

Comments and Suggestions for Authors

This study investigated the automated prediction of consolidation tumor ratio for stage I NSCLC from treatment planning CT images, based on deep learning segmentation models. The reviewer has the following concerns.

(1)The biggest issue is that the data from  treatment planning CT instead of diagnostic CT, which may affects the accurate of the volume of tumor, due to the lack of breath holding, there will be respiratory motion artifacts, which will lead to image blurring.

(2)For the CT imaging, the slice thickness range is too large( from 0.625mm~5mm), which can affect the results.

(3)In table 2, what is the number for the 3 different histology?

(4)In the method section, how is the patient's survival time obtained? It suddenly appears in the part of results, which is very weird.

(5)What kind of therapy did the patients accept? Different patients undergo different treatment, which affect their survival . Therefore, the statistics on the relationship between semi-solid nodules and survival may be different.

(6) The study is only focus on part solid tumor, but there are solid and pure GGO patients also enrolled, which is not related with the study.

Comments on the Quality of English Language

The English writing is good, and the paper is well organized.  However, do not use the first person in writing.

Round 2

Reviewer 1 Report

Comments and Suggestions for Authors

All the comments and suggestions are carefully addressed in the revised manuscript. The effort of the authors is highly appreciated.

It is believed that the readers of the journal can get valuable insights into the research question, proposed solution, and the corresponding evaluation based on the manuscript.

Author Response

Thank you for your suggestion, it has been of great help to us。

Reviewer 2 Report

Comments and Suggestions for Authors

The authors have proposed the responses to my comments one by one. However, I also have a few more suggestions for the revised manuscript.

1.  Suggest adding non-small cell lung cancer (NSCLC) as a keyword.

2. The quality of some figures should be improved, especially Fig.8, Fig.11, Fig.13.

Reviewer 3 Report

Comments and Suggestions for Authors

The authors implemented all the commentaries suggested and the new version of the paper is more complete and clearer. For this situation I recommend  its publication in Applied Science Journal.

Author Response

Thank you for your suggestion, it has been of great help to us.

Reviewer 4 Report

Comments and Suggestions for Authors

This version is better than the previous one. But the reply to comment 4 is same with comments 3, so please response to comment 4. Thanks. 

Comments on the Quality of English Language

Please avoid using the first person in the writing.
